# Sweat Chloride Testing and Nasal Potential Difference (NPD) Are Primary Outcome Parameters in Treatment with Cystic Fibrosis Transmembrane Conductance Regulator (CFTR) Modulators

**DOI:** 10.3390/jpm11080729

**Published:** 2021-07-27

**Authors:** Isabelle Sermet-Gaudelus, Thao Nguyen-Khoa, Aurélie Hatton, Kate Hayes, Iwona Pranke

**Affiliations:** 1INSERM U1151, Institut Necker Enfants Malades, Université Paris Sorbonne, 75015 Paris, France; thao.nguyen-khoa@aphp.fr (T.N.-K.); aurelie.hatton@inserm.fr (A.H.); iwona.pranke@inserm.fr (I.P.); 2Service de Pneumologie et Allergologie Pédiatriques, Centre de Référence Maladie Rare Mucoviscidose et Maladies Apparentées, Hôpital Necker Enfants Malades, AP-HP Centre Université de Paris, 149 rue de Sèvres, 75743 Paris, France; 3European Cystic Fibrosis Society-Clinical Trial Network, 7470 Karup, Denmark; K.Hayes@qub.ac.uk; 4Lung, European Reference Network, 75743 Paris, France; 5Laboratoires de Biochimie et du Centre Régional de Dépistage Néonatal, Hôpital Necker Enfants Malades, AP-HP Centre Université de Paris, 75015 Paris, France; 6Northern Ireland Clinical Research Facility (NICRF), Wellcome-Wolfson Institute for Experimental Medicine, Belfast City Hospital & Queen’s University Belfast, Belfast BT9 7AB, UK

**Keywords:** sweat chloride test, nasal potential difference, NPD, CFTR

## Abstract

With the advent of CFTR modulators, surrogate outcome parameters that accurately quantify the improvement in CFTR activity are needed. In vivo biomarkers that reflect CFTR ion transport and can serve as outcomes in the treatment of CFTR modulators are the sweat Cl^−^ test (SCT), the nasal potential difference (NPD) measurement or the intestinal current measurement (ICM). This review focus on the SCT and NPD. The SCT displays a low intra-patient variability in contrast to the NPD. It has been used extensively as a biomarker of CFTR function in clinical trials of CFTR modulator therapies and provides evidence for change in the short term. The level of functional rescue in the NPD increases up to 40% of normal CFTR in patients with a Gly551Asp treated with ivacaftor monotherapy, while in F508del homozygous patients treated with ivacaftor-lumacaftor, activity increased on average up to ~20% of normal activity. While both tests provide evidence of the effect on CFTR activity, they cannot be used at an individual level to predict the response to any CFTR modulators. Nevertheless, their rapid modification, reflecting electrophysiological properties, highlight their potential use in proof-of-concept studies for CFTR modulators.

## 1. Introduction

Cystic Fibrosis Transmembrane Conductance Regulator (CFTR) is a membrane protein and an anion channel located at the apical membrane of epithelia with a high expression in the airways and the ducts of exocrine glands such as the sweat gland, the pancreas or the *vas deferens* [1]. Mutations in the *CFTR* gene are associated with abnormal production and/or function of the CFTR protein. This results in defective chloride (Cl^−^), bicarbonate ions, sodium (Na^+^) and water transport and production of abnormal respiratory and pancreatic secretions, with high Cl^−^ levels in sweat. More than 2000 different variants in the *CFTR* gene have been reported. The most frequent is F508del and is carried by more than 80% of people with Cystic Fibrosis (pwCF) on at least one allele. In each country, only five or six other mutations are found in more than 1% of pwCF, the other mutations being very rare [2].

More than 30 years after the discovery of the CFTR gene, therapeutic strategies of CF have been reshuffled by the advent of proteic therapies targeting mutated proteins [3,4]. Those CFTR modulator drugs restore CFTR function, and this is associated with amazing improvement in clinical status of patients carrying the F508del mutation as well as those with rarer mutations which affect gating, conductance or protein folding [5]. Four molecules are currently being used in patients alone or in combination: ivacaftor, lumacaftor/ivacaftor, tezacaftor/ivacaftor and elexacaftor/tezacaftor/ivacaftor. However, the response of patients is specific to the CFTR mutation. Moreover, among patients with a similar mutation, the response can be variable as well, the mechanism of which is still unclear. Therefore, to prove or more optimally predict the effect of the treatment is more important than ever. This highlights the need for surrogate outcome parameters that accurately quantify the improvement in the underlying disease defect. These include in vivo CFTR functional assays which measure mainly CFTR-mediated Cl^−^ transport and its change upon CFTR modulators in the short term, as surrogate outcomes of the pharmacological response in clinical trials of CFTR protein modulators [6]. They are mainly used to enlarge indications for rare or ultra-rare variants, to assess new corrector efficacy in proof-of-concept trials or as confirmation in larger phase 3 trials. They might be also used in precision medicine to select the optimal treatment needed by a given individual among different drugs or drug combinations [7]. In vivo biomarkers that reflect CFTR ion transport and can serve as outcomes in the treatment of CFTR modulators are the sweat Cl^−^ test (SCT), the nasal potential difference (NPD) measurement or the intestinal current measurement (ICM). Otherwise, an ex vivo biomarker that reflects CFTR-mediated Cl^−^ secretion is the short-circuit current measurement in patients’ derived nasal epithelial cells (HNEs) [8,9,10]. This is based on the fact that the correlation of CFTR activity in nasal cells between in vivo and ex vivo measurements according to the genotype, in a large group of rare mutations, is significant [8]. This review will focus on the SCT and NPD.

## 2. Sweat Test

### 2.1. Physiological Principles and Relation to CFTR Activity

Reduced CFTR function in the sweat duct decreases Cl^−^ and Na^+^ reabsorption and potentiates water loss, resulting in elevated sweat Cl^−^ concentrations. One of the main advantages of the SCT is its low cost, non-invasiveness, practicality for trained staff and its standardisation by different Standard Operating Procedure (SOP) guidelines [11,12,13,14]. After 30 min of stimulation of the sweat gland with pilocarpine by iontophoresis, the sweat is collected with the Macroduct^®^ collection system and the Cl^−^ concentration is measured by coulometry or direct potentiometry for the optimal methods [15]. The SCT differentiates CF subjects and healthy controls with validated thresholds, e.g., normal values < 30 mmol/L Cl^−^, intermediate values between 30 and 59 mmol/L Cl^−^ and abnormal values ≥ 60 mmol/L Cl^−^, at any age [13,14,15]. There are very few limitations for this test, mainly infants before the age of 3 days, <36 weeks corrected gestational age or weight < 2 kg [11,12,13,14,15]. Malnutrition, hydration imbalance, skin problems, metabolic diseases (fucoscidosis, glycogenosis and muco-polysacharidosis), untreated hypothyroidism, pseudo-hypoaldosteronism, hypoparathyroidism, panhypopituitarism, renal insufficiency, oedema and some medications (such as systemic corticoid treatment, prostaglandin and sodium chloride perfusion) can lead to false-positive or false-negative sweat tests [11,12,13,14,15].

Interestingly, McCague et al. showed that CFTR function assessed in vitro in cell lines transfected with mutant CFTR and the sweat Cl^−^ concentration of individuals with CF carrying these genotypes were correlated with a logarithmic relationship [16]. This nonlinear relationship may be explained by the fact that a small fraction of “normally functioning” CFTR channels “may be sufficient” to restore “close-to-normal” sweat Cl^−^ level. This trend to normalisation of the microenvironment in the membrane vicinity might also impact function of other transporters involved in trans-epithelial fluid transport. As there is no secondary damage to the sweat gland in CF [17,18], it can be hypothesised that the SCT results of patients upon modulators can be compared to patients with milder genotypes and, thus, serve as a proxy of phenotype. For example, McCague et al. showed that an SCT below 70 mmol/L Cl^−^ is associated with a CFTR function above 10% and a ppFEV_1_ measurement above 80% into adulthood [16]. Thus, this threshold of 10% might be the benchmarking for CFTR modulation therapy. Moreover, this logarithmic relationship implies that small increases in CFTR function (2 to 5%) could generate substantial sweat Cl^−^ increase and clinically relevant lung function improvement. This parallels observations from individuals carrying the G551D variant, in which moderate increases in the SCT resulted in remarkable improvement in CF clinical measures [19].

### 2.2. Analytical and Biological Variability

Studies are still lacking in the assessment of the SCT in patient-to-patient variability. Vermeulen et al. studied patients with 2 CF-causing mutations, e.g., within test repeatability (difference between samples from right and left arm on the same test occasion) and found a range between −3.2 and +3.6 mmol/L Cl^−^, while the between test limits of repeatability (difference between results from two tests occasions) were between −18 and +14 mmol/L Cl^−^ [19,20]. The variability has been less studied for patients with an intermediate SCT. Focussing on these patients, Vermeulen et al. showed that patients translating below the threshold of 30 mmol/L Cl^−^ at the second test (median interval 3.5 weeks) were younger and had a lower initial SCT, whilst when the second test remained in the intermediate range, the patients were more at risk to carry CFTR mutations [20]. Similar results were reported by Cirilli et al., whereby CF patients displayed the lowest variability (CV% values 20.2) *versus* 31.1 in healthy subjects [21]. In a study of 1761 twins/siblings with CF, the main factors of variability, apart from the type of mutation, which represented more than 50% of the variability, included time of sampling, environmental factors, e.g., climate and family diet, but also individual unknown factors, e.g., modifier genes or environmental exposures on the day of the sampling [22].

In clinical trials, the variability should be smaller due to inclusion of only CF patients and the use of standardised collection techniques and centralised laboratory practices. LeGrys et al. showed that the analytical variation was limited to 4%, the intra-individual variation and the inter-individual variation were below 6% and 9% respectively, which is small [23]. The variability was not affected by gender, collection site or sample weight. As a whole, the variance (SD) within the test was 4.8 mmol/L Cl^−^, for inter-individuals 8.9 mmol/L Cl^−^, and for intra-individuals 8.1 mmol/L Cl^−^, which provides remarkable test performance.

These observations have different consequences in clinical research. First, averaging the Cl^−^ concentrations at the two sample sites is better than selecting one site based on the greater sweat weight or the Cl^−^ concentration [23]. Second, a diagnostic historical SCT can be used as a reliable estimate [23]. Indeed, the difference between historic and enrolment sweat Cl^−^ values was very small in different clinical trials: mean change −1.2 mmol/L Cl^−^ (95% CI [−3.7;1.3] mmol/L Cl^−^) over an average 16.2 years in the G551D-CFTR GOAL cohort, 0.5 mmol/L Cl^−^ (95% CI [−2.4;3.4] mmol/L Cl^−^) over an average of 18.9 years in the homozygous F508del-CFTR PROSPECT cohort and 0.9 mmol/L (95% CI [−3.3;5.2] mmol/L Cl^−^) over an average of 19.4 years in the heterozygous F508del-CFTR. Thus, the SCT is a highly robust test to be used in phase 2 trials. The only exception is within the infant population. Collaco et al. showed a higher within-subject variation in this population [22]. This might be due to an increase in the SCT during infancy as shown by Legrys et al., who reported a statistically significant increase from early (89.8 mmol/L Cl^−^) to late (95.0 mmol/L Cl^−^) infancy, and more specifically from 99 mmol/L Cl^−^ in F508del-CFTR homozygotes under 1 year up to a mean of 106 mmol/L Cl^−^ in those over 3 years of age [23]. The within-subject variability of SCT measurements has also been observed for CRMS (CFTR-related metabolic syndrome) and CFSPID (CF screen positive inconclusive diagnosis) subjects, known as CF screen positive patients [24]. Although those changes are not clinically significant, these observations highlight the need to perform prospective studies to determine the variability of the SCT results in infants with CF for the future design of clinical trials in young patients.

### 2.3. Sweat Test as a Primary Outcome in Clinical Trials with CFTR Modulator Therapies

The SCT has been used extensively as a biomarker of CFTR function in clinical trials of CFTR modulator therapies [17,19,23,24,25,26]. This was inaugurated by the first amazing initial proof-of-concept studies on ivacaftor in G551D patients, which demonstrated improvement in sweat Cl^−^ to near-normal values [27].

At the statistical level, the SCT has proved to be a robust pharmacological biomarker across clinical trials [6,17,25]. As a whole, average decreases in sweat Cl^−^ have correlated with lung function improvement [26,27,28,29]. In a *post hoc* analysis, Fidler et al. pooled the data from multiple cohorts of patients treated for at least 14 days with ivacaftor monotherapy [28]. There was a significant correlation between SCT levels and ppFEV_1_ changes, but this was not confirmed at the individual level [30]. This is probably due to the fact that the ppFEV_1_ improvement over months may be jeopardised by structural damage to the airways and that other outcome parameters, such as distal obstruction parameters, lung clearance index, lung imaging or lung function decline over time, should be considered.

Within-individual variability in SCT values following ivacaftor treatment and other CFTR modulators was stable below 10% [5,14,31,32]. Therefore, at the individual level, changes in the SCT of at least 10% are beyond inherent analytical and biological variation and should represent physiologically significant differences in CFTR activity in sweat Cl^−^. For example, Legrys et al. reported that a power of 90% is anticipated for a drug decreasing the SCT by at least 15 mmol/L Cl^−^, with a design of four bilateral measurements in each of the periods of a crossover trial including five patients. Importantly, considering the fact that McCague et al. related sweat Cl^−^ level to lung function, it should be more relevant to consider the absolute post treatment SCT values rather than the change upon treatment, to evaluate CFTR functional restoration and prediction of long-term outcome [16].

Most interestingly, for highly efficient modulators (ivacaftor, elexacaftor/tezacaftor/ivacaftor (Kaftrio^®^)) the variation seems to occur very early, as early as 15 days, and remain stable across time from 1 to 6 months post-modulator after initiation of therapy [5,31,32,33]. This may be even as early as 3 days, as illustrated in Figure 1. This observation highlights the potential to use this biomarker in very short-term proof-of-concept trials.

In real life studies in a post-approval setting, sustained reductions were shown during the first months of Orkambi^®^ (lumacaftor/ivacaftor), demonstrating a strong correlation between changes at 1 month and 6 months of treatment [34,35,36,37].

## 3. Nasal Potential Difference

### 3.1. Physiological Principles and Relation to CFTR Activity

The NPD is an in vivo test which indirectly reflects trans-epithelial transport across the nasal epithelium by quantifying the resulting voltage change. It assesses the function of the epithelial sodium channel (ENaC), an Na^+^ transporter, which is inhibited by Amiloride and the total Cl^−^ transport as well as its more specific activation, CFTR-related, by beta-agonists. Importantly, it is a proxy of the lower airway trans-epithelial transport as at 2–3 cm distal to the nostril, the nasal epithelium displays a pseudostratified columnar epithelium similar to that of the distal airways.

CFTR is highly expressed in the apical membranes of respiratory epithelial cells [38]. The trans-epithelial ion transport between the two compartments (blood/basal and lumen airway/apical) results in a polarisation of the membrane (negative on the intracellular side and positive on the extracellular side), creating a transmembrane electric potential. This is measured using a high impedance voltmeter by assessing the voltage difference between that of an electrode placed via a conducting bridge in contact with the mucosa of the nasal epithelium along the inferior turbinate (exploring electrode, outside the cells) and the other placed in the subcutaneous compartment in the forearm (the reference electrode, inside the cells). SOP protocols are available and aim to harmonise the protocols in all CF sites [39].

As airway epithelium displays a Na^+^ absorption at the basal state, the basal potential is usually negative, with values between −15 and −25 mV in healthy controls. In CF patients, the basal potential is much more negative (−35 to −50 mV) due to the absence of inhibition of ENaC by CFTR, leading to excessive Na^+^ absorption. After perfusion of Amiloride, there is a decrease in absorption of Na^+^, leading to a depolarisation of the membrane which is greater in CF due to a more negative basal potential. Sequential perfusion of the nasal mucosa by a solution not containing Cl^−^ creates a chemical gradient for Cl^−^ secretion via Cl^−^ channels, including CFTR. In order to balance the membrane electric potential, Cl^−^ is secreted out of the cell through the CFTR channel, causing a repolarisation of the potential. By adding β2-adrenergic drug isoprenaline, CFTR is specifically activated, which increases the repolarisation. In CF patients, there is a lack of response and, thus, no repolarisation. Finally, a solution containing ATP is perfused, which increases the repolarisation in both CF and non-CF epithelia through CFTR-independent calcium-dependent ion channels (Figure 2). The main parameters in the NPD measurement include the response to Amiloride perfusion (ΔAmiloride PD), the response to zero-chloride and isoprenaline perfusion (ΔCl^−^free + isoprenaline PD). Validated scores can combine the total Cl^−^ response (TCR) and the Amiloride response.

The advantages of NPD testing are that it measures ionic transport of the airway epithelium in vivo, reflecting one of the triggers of the pathophysiological cascade of CF lung disease. Moreover, CFTR and ENaC activity can be differentially assessed. Most importantly, this in vivo evaluation is specifically correlated to in vitro evaluation of CFTR in the cells of the same patients, as is shown in Figure 3.

Its drawbacks lie in the need for a sophisticated infrastructure, trained technicians and a pharmacy able to produce the perfusion solutions. Most importantly, intact nasal mucosa is required to obtain adequate measurements (e.g., smokers or patients with chronic rhinitis, who can have very flat tracings). The cooperation of the patients is also required; hence, it may be difficult to perform in infants.

Importantly, we showed that in vivo NPD measurements of CFTR activity significantly correlated with CFTR activity measured in vitro on HNE cells of the same patient, including patients with rare mutations. Concordantly, a significant correlation between CFTR function and apical CFTR expression in HNE cells was observed [8]. An evaluation of CFTR function correction in HNE cells upon Orkambi^®^ treatment allowed consideration of the measurements of short-circuit current on HNE cells as a surrogate outcome, since it significantly correlated with FEV_1_ improvement in the same F508del-homozygous patients after 6 months of treatment [8,9]. Similar published and unpublished reports by the author support this concept [10].

### 3.2. Analytical and Biological Variability

The NPD differentiates patients with CF from healthy controls [39,40]. Patients with mild phenotypes show intermediate NPD values reflecting the residual CFTR function.

The main challenge with the NPD is the poor repeatability of the measurements. Studying the placebo arm of a multicentre study, Kirilli et al. showed an intrinsic variability, with basal PD and ΔAmiloride displaying the highest variabilities, mainly stemming from inter-centre effect, thus highlighting the operator-dependent aspect of these measurements [41]. ΔLowCl^−^, ΔIsoproterenol and ΔLowCl^−^-Isoproterenol demonstrated a large intra-subject variability within ± 7.2 mV. It was greater in patients reporting ongoing pulmonary exacerbations, which seems to be a factor influencing variability. Other factors including environmental pollution and possibly more subtle physiological variations, such as oestrogen fluctuation during menstrual variation may also influence variability. Attempts to improve repeatability demonstrated that warming the solutions did not significantly modify the response and that use of the large surface catheter at a fixed location might be considered beneficial [42,43].

## 4. Clinical Trials

CFTR-mediated Cl^−^ transport across nasal epithelium appears to exhibit a nonlinear relationship with the SCT [Cl2] [14,42]. Proof-of-concept studies with ivacaftor showed a dose-response progressive improvement [6], both for NPD endpoints, sweat test and ppFEV_1_ [43].

The NPD has also been used as an exploratory endpoint in real life trials, measuring the change in CFTR function during treatment with lumacaftor-ivacaftor-treated patients [35,36].

The level of functional rescue in patients with a Gly551Asp with ivacaftor monotherapy resulted in improvements to 43% of normal CFTR activity in the NPD and to 52% in ICM studies, respectively [44,45], while F508del-CFTR activity increased on average up to ~20% of normal in the nasal epithelium and ~15% in the rectal mucosa, reaching levels observed in subjects with genotypes associated with residual function [36,37]. This range of CFTR activity is comparable to that observed for residual CFTR-mediated ion transport in the NPD and ICM of patients with mild CF and with long-term exocrine pancreatic sufficiency [39,45].

Importantly, in both studies, the NPD change did not correlate to ppFEV_1_ change, indicating that recovery in CFTR function was not related to clinical benefit. Moreover, there was no correlation in the level of rescue of CFTR activity between the biomarkers, indicating that they are not interchangeable. This can be explained by the fact that the two tests reflect CFTR activity differently: the sweat test assesses Cl^−^ reabsorption along the sweat duct, while the NPD measures trans-epithelial voltage indirectly, by relative changes in ion conductance. Moreover, the bioavailability of the drug may differ in the three epithelia.

## 5. Conclusions

Both the SCT and NPD can be primary outcome parameters in treatment with cystic fibrosis transmembrane conductance regulator (CFTR) modulators. While both provide evidence of the effect on CFTR Cl^−^ transport activity, they cannot be used at an individual level to predict the response to any CFTR modulators. Nevertheless, their rapid modification, reflecting electrophysiological properties, highlight their potential use in proof-of-concept studies for CFTR modulators.

## Figures and Tables

**Figure 1 jpm-11-00729-f001:**
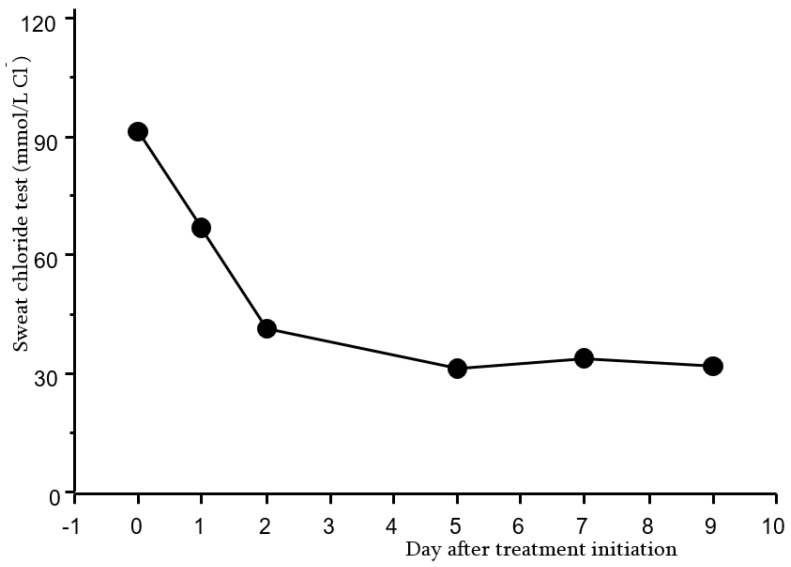
Variation of the sweat chloride concentration after Kaftrio^®^ in a F508del homozygous patient.

**Figure 2 jpm-11-00729-f002:**
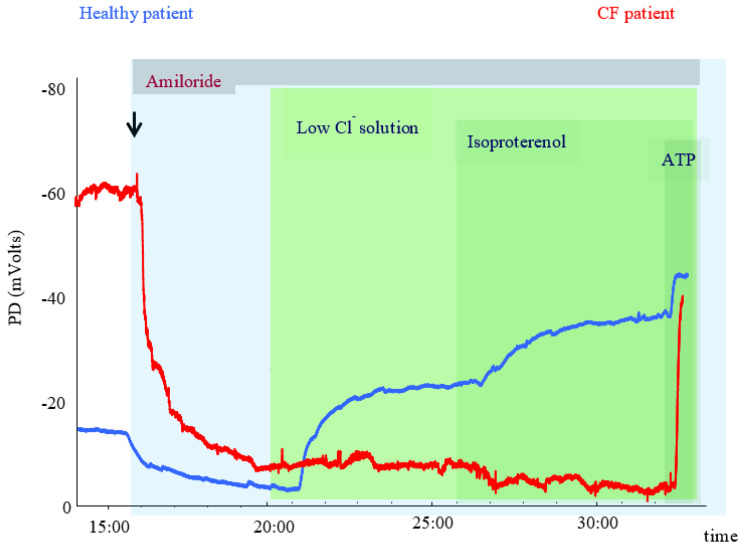
NPD tracing in a healthy control and in an F508del homozygous patient.

**Figure 3 jpm-11-00729-f003:**
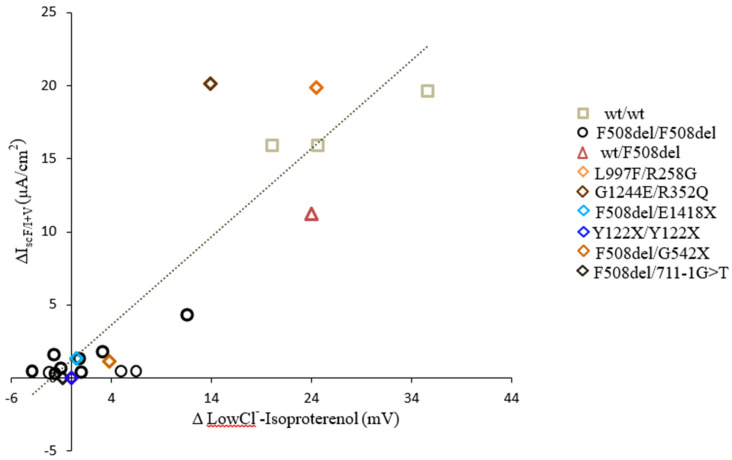
Correlation between cAMP-activated chloride transport in the nasal mucosa by the NPD (∆ LowCl^−^-Isoproterenol (mV) and in the primary cell culture from nasal brushing (∆I_sc F/I+V_ (µA/cm^2^) in the same patient.

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
