# Peer review of "Sweat Chloride Testing and Nasal Potential Difference (NPD) Are Primary Outcome Parameters in Treatment with Cystic Fibrosis Transmembrane Conductance Regulator (CFTR) Modulators"

_jpm, 2021, doi:10.3390/jpm11080729_

Round 1
Reviewer 1 Report
This is a very interesting article and it was a pleasure for me to review it. The authors focus on sweat chloride test and nasal potential difference as primary outcome for therapy with CFTR modulators.
In my opinion some changes could be useful:
- As already written, functional tests to predict the effect of CFTR modulators in CF patients are crucial, in particular for patients with rare CFTR mutations, so not elegible for clinical trials. Authors should also talk about the usefulness of nasal brushing for this purpose. There are several papers about this topic or case report such as this: Terlizzi V, et al. Ex vivo model predicted in vivo efficacy of CFTR modulator therapy in a child with rare genotype. Mol Genet Genomic Med. 2021 Apr;9(4):e1656.
- The chapter about the variability of the sweat test is very interesting. This variability is especially known in infants and even more so in CRMS/CFSPID subjects. I think that it could be useful mention this aspect too.
In general the work is done well; it also explains the limitations of such tests. I would be careful to point out doubts about the link between sweat testing and respiratory function. I also believe that in any case the sweat test is more reliable than NPD as an outcome (they do not have the same value as written in the conclusions)
Author Response
- “As already written, functional tests to predict the effect of CFTR modulators in CF patients are crucial, in particular for patients with rare CFTR mutations, so not elegible for clinical trials. Authors should also talk about the usefulness of nasal brushing for this purpose. There are several papers about this topic or case report such as this: Terlizzi V, et al. Ex vivo model predicted in vivo efficacy of CFTR modulator therapy in a child with rare genotype. Mol Genet Genomic Med. 2021 Apr;9(4):e1656.”
We thank the Reviewer for this important suggestion.
Indeed we published already 3 studies on the topic. We first demonstrated the correlation on this model between the expression of the CFTR protein at the membrane and its activity for a number of mutations (Pranke et al.. Am J Respir Crit Care Med 2019;199(1):123-126; and Sci Rep 2017 7(1):7375). We also recently demonstrated the correlation between the activity of CFTR in this model and in the native cells by the patch clamp method (Noel et al, in review at J Physiol). We showed, in a pilot study, that the level of CFTR functional restoration in primary cells derived from nasal brushing upon incubation with CFTR modulators, was correlated to the improvement in respiratory clinical status. A French collaborative study including 100 patients treated with Orkambi® and correlating the in vitro effect on CFTR activity on the nasal primary cells with the effect on FEV1 at 6 months is ongoing , as well as a similar project in patients with rare variants. The text : “Otherwise, an ex vivo biomarker that reflects CFTR-mediated Cl- secretion is the short-circuit current measurement in patients’ derived nasal epithelial cells (HNEs) [8-10]. This is based on the fact that the correlation of CFTR activity in nasal cells between in vivo and ex vivo measurements according to the genotype, in a large group of rare mutations, is significant [8]..” has been added at the end of the Introduction part together with the references (page 4). And …: “Importantly, we showed that in vivo NPD measurements of CFTR activity significantly correlated with CFTR activity measured in vitro on HNE cells of the same patient, including patients with multiple genotypes with rare mutations. Similarly, a significant correlation between CFTR function and apical CFTR expression in HNE cells was observed for rare mutations-genotypes [8]. An evaluation of CFTR function correction in HNE cells upon Orkambi® treatment allowed consideration of the measurements of short-circuit current on HNE cells as a surrogate outcome, since it significantly correlated with FEV1 improvement in the same F508del-homozygous patients after 6 months of treatment [8,9]. Similar published and author unpublished reports support this concept [10].” has also been added (pages 12-13).
- “The chapter about the variability of the sweat test is very interesting. This variability is especially known in infants and even more so in CRMS/CFSPID subjects. I think that it could be useful mention this aspect too.”
- We thank the Reviewer. This information has been included. We added the phrase: “The within-subject variability of SCT measurements has also been observed for CRMS (CFTR-related metabolic syndrome) and CFSPID (CF screen positive inconclusive diagnosis) subjects, known as CF screen positive patient [24].”We added the referenced publication indicating a direct correlation between age and sweat Cl in patients with CRMS in the Analytical and biological variability section (page 7).

Reviewer 2 Report
Thank you very much for this very interessing manuscript.
Page 4, second-last line: remove "hypoparathyroidy" and replica with"hypoparathyroidysm"
Discussion, second-last line: remove ","
Author Response
Reviewer 2
- “Page 4, second-last line: remove "hypoparathyroidy" and replica with"hypoparathyroidysm"”
- We thank the Reviewer for this remark. "hypoparathyroidy" has been changed into "hypoparathyroidysm".
- “Discussion, second-last line: remove ","”
- The “,” sign has been removed.